# Valorization of Rice Husk and Straw Agriculture Wastes of Eastern Saudi Arabia: Production of Bio-Based Silica, Lignocellulose, and Activated Carbon

**DOI:** 10.3390/ma15113746

**Published:** 2022-05-24

**Authors:** Hisham S. M. Abd-Rabboh, Khaled F. Fawy, Mohamed S. Hamdy, SeragEldin I. Elbehairi, Ali A. Shati, Mohammad Y. Alfaifi, Hala A. Ibrahium, Saad Alamri, Nasser S. Awwad

**Affiliations:** 1Department of Chemistry, Faculty of Science, King Khalid University, P.O. Box 9004, Abha 61413, Saudi Arabia; khossayn@kku.edu.sa (K.F.F.); mhsaad@kku.edu.sa (M.S.H.); aawwad@kku.edu.sa (N.S.A.); 2Department of Biology, Faculty of Science, King Khalid University, P.O. Box 9004, Abha 61413, Saudi Arabia; aaalshati@kku.edu.sa (A.A.S.); alfaifi@kku.edu.sa (M.Y.A.); habrahem@kku.edu.sa (H.A.I.); saralomari@kku.edu.sa (S.A.)

**Keywords:** rice husk, rice straw, silica, lignocellulose, activated carbon, agriculture waste recycling

## Abstract

Bio-based silica, lignocellulose, and activated carbon were simply produced via the recycling of Hassawi rice biomass waste of Al-Ahsa governorate in the eastern Saudi Arabia region using a fast chemical treatment procedure. Rice husk and rice straw wastes were collected, ground, and chemically treated with sodium hydroxide to extract silica/silicate from the dried plant tissues. The liquid extract is then treated with acid solutions in order to precipitate silica/silicate at neutral medium. Lowering the pH of the supernatant to 2 resulted in the precipitation of lignocellulose. Thermal treatment of the biomass residue under N_2_ gas stream resulted in activated carbon production. Separated products were dried/treated and characterized using several physical examination techniques, such as FT-IR, SEM/EDX, XRD, and Raman spectroscopy in order to study their structure and morphology. Silica and lignocelluloses products were then preliminarily used in the treatment of wastewaters and water-desalination processes.

## 1. Introduction

Rice is considered one of the most important and most consumed cereal crops in the diet of the people of Saudi Arabia, with an average consumption rate of 40 kg per person per year [1]. Although a large percentage of consumed rice in KSA is imported, its cultivation in the eastern region of the kingdom (Al-Hasa area) is improving very fast. Oryza sativa L. is the type of rice that is grown in this area, with characteristic properties of brown color, drought resistance, and salty soil resistance, and is known as Hassawi rice [2,3,4].

Rice production is always associated with serious environmental threats concerned with the huge biomass of agricultural waste formed as side products of the production process. The average world production of rice is in the range of 760 million tons per year, with an approximate rice straw waste of 1140 million tons [5]. About half of that biomass waste is burnt in many countries all over the world, emitting vast amounts of organic and inorganic air pollutants, increasing global warming and resulting in serious environmental problems [6,7].

Researchers have found out that this huge amount of biomass is rich in a variety of important natural materials, such as silica and lignocellulose [8,9,10]. Recent research established that natural mesoporous silica nanoparticles (MSN) extracted from rice and wheat husks were known to be good natural biogenic carriers for drug delivery in cancer treatment [11], and were successfully applied in the fields of catalysis [12,13], enzyme immobilization [14], and treatment/removal of inorganic, organic, and gaseous species [15]. MSN is simply produced from sodium silicate solutions obtained through alkaline chemical treatment of husk waste [16,17].

Silica aerogel, with excellent properties of high porosity and surface, has low thermal conductivity and low density, being produced from rice straw through a combustion/freeze-drying process [18]. An efficient complex of zinc with lignin, silica, and fatty acids natural materials was produced from rice straw and was successfully used as an antioxidant and as an activator in the fabrication of rubber composites [19]. A nanocomposite of silica/CaCO_3_ was obtained through the combustion of rice straw and was used as filler in paper production [20]. Microsilica particles extracted from rice straw was effectively used as an enhancing material for the mechanical properties of pavement quality concrete [21]. Nanosilica was also produced from rice straw through a simple and efficient carbonization-extraction procedure [22].

Lignocellulose is one of the main components of rice husk, which is extracted after treating the biomass with alkaline solutions followed by precipitation/removal of silica in a neutral medium, then treating the filtrate with ionic liquids/acids to separate the lignocellulose [23]. Studies have focused on the extraction of lignocellulose from a large number of rice straw samples harvested under different agricultural and weather conditions [24].

Another important material that is produced from rice straw and husk biomass is activated carbon (AC), a non-graphitic-graphitizable carbon with much-disordered microstructure [18]. AC is characterized by both high porosity and high surface area besides its thermal stability, which enhanced its extensive use in the fields of water purification and wastewater treatment [19,20]. In addition, due its electrical properties, it is used in the industry in capacitors, batteries, energy-storage, and in catalyst production [21]. Additionally, it is widely used for the removal of inorganic and organic species, as well as carbon dioxide and other environmentally harmful gases that cause global warming [22,23,24,25,26,27]. AC is obtained from different sources, and recent approaches concentrate on producing AC via recycling wastes of industrial and agricultural origin [28]. Rice husk and straw wastes are considered rich sources of AC with distinguished properties [29,30,31]. The obtained carbon is then activated via a physical or a chemical process. Physical activation is conducted by igniting the product under CO_2_ or N_2_ gas streams [32], while chemical activation involves treating the material with chemicals such as sodium hydroxide (NaOH), phosphoric acid (H_3_PO_4_), and zinc chloride (ZnCl_2_). Chemically AC is found to have better properties in terms of good mesoporous structure and large surface area [33,34].

In this article, the authors introduce a simple chemical treatment method for the recycling of Hassawi rice husk and straw agricultural waste gathered from the Al-Hasa region in eastern Saudi Arabia, in order to produce several natural important materials including silica, lignocellulose, and activated carbon. These promising low-cost and environmental friendly natural materials will be applied in the fields of bio-fuel production, water desalination, energy storage, and anticancer studies.

## 2. Materials and Methods

### 2.1. Materials and Reagents

Sodium hydroxide (NaOH), potassium hydroxide (KOH), hydrochloric acid (HCl), pluronic F-127 copolymer, sulfuric acid (H_2_SO_4_), potassium permanganate (KMnO_4_), hydrogen peroxide (H_2_O_2_), and sodium borohydride (NaBH_4_) were obtained from Sigma-Aldrich, St. Louis, MO, USA (www.sigmaaldrich.com, accessed on 30 January 2022). High purity nitrogen gas (N_2_, 99.999%) was used for the activation process. All used chemicals were of analytical grade.

### 2.2. Equipment

Samples were dried at 80–105°C inside a Binder-drying-oven; Model ED 56, Germany (Binder GmbH, Tuttlingen, TU, Germany). The carbonization process was performed in an MTI-three-zone tube furnace; model OTF-1200X-III-S (MTI Co., Richmond, CA, USA). A programmable muffle furnace was used for sample ignition.

### 2.3. Extraction and Treatment Procedures

All solutions were prepared using deionized water. The extracting solution was prepared by dissolving 300 g of NaOH in 1 L water. The solution volume was then brought up to 6 L to produce 50 g/L NaOH solution. Both rice husk and rice straw wastes (250 g portions) were treated separately with 3 L of the NaOH extracting solution. The mixtures were boiled for 30 min with constant stirring and then left to stand at room temperature overnight. Residues were then filtered off and supernatants were collected. The pH’s of the collected filtrates were adjusted down to pH = 7 using concentrated HCl solution, at room temperature with constant stirring, in order to precipitate silica/silicate. The separated precipitates were filtered off, washed thoroughly with water, and dried overnight in a drying oven at 105°C. Dried samples were then ignited at 650 °C for 2 h (with a temperature ramp of 120 °C/h) in order to burn all organic matter.

Filtrates of the last step, with pH = 7, were further treated with concentrated HCl solution to lower their pH values down to 1 in order to precipitate lignocellulose. Precipitated material produced from the rice husk sample was collected, washed with water, and dried in a drying oven at 105 °C for 8 h, followed by carbonization under N_2_/CO_2_ gas flow at 800 °C in a tube furnace oven for 8 h. On the other hand, precipitated material produced from the rice straw sample was collected, washed with water, and dried at 60°C in a drying oven for 24 h. The dried sample was then divided into two portions: one was kept as is, and the other was subjected to carbonization under N_2_/CO_2_ gas flow at 800°C in a tube furnace for 6 h.

### 2.4. Sample Characterization

Raw materials, intermediate extracts, and final products were physically characterized using the following instruments: Diamond ATR Fourier-transform Infrared spectrometer, ATR-FT-IR (Agilent, Cary 630, Santa Clara, CA, USA, www.agilent.com, accessed on 30 January 2022) for functional group analysis; X-ray Diffractometer with 2θ in the range 10–80° and 0.03° rate (Shimadzu, XRD6000, Kyoto, Japan, www.shimadzu.com, accessed on 30 January 2022) for crystallographic studies and phase analysis; a field-emission scanning electron microscope equipped with X-ray energy dispersion, SEM/EDX (Jeol, L6340, Kyoto, Japan, www.jeol.com, accessed on 30 January 2022) for surface morphology and composition analysis; BET surface analysis (QuantaChrome, NOVA 2000e, Boynton Beach, FL, USA, www.quantachrome.com, accessed on 30 January 2022) for pore size distribution and N_2_ sorption studies.

### 2.5. The Conversion of Sodium Silicate to Amorphous Silica

The method reported by Kosuge et al. was adopted to prepare porous amorphous silica from the bio-based sodium silicate, which was obtained from rice straw/husk [35] by using pluronic as a triblock copolymer. In a typical synthesis process, 2 g of pluronic F-127 (98%, Sigma Aldrich, St. Louis, MO, USA) was dissolved in 80 g of an aqueous solution of HNO_3_ at room temperature. This solution was added to another aqueous solution of sodium silicate (5 g in 15 mL of deionized water). The formed mixture was vigorously stirred at room temperature with a stirring rate of 700 rpm for 2 h. After elapsing the stirring time, the formed turbid solution was filtered and washed by using deionized warm water. Then the obtained solid product was dried at 98°C for 4 h, followed by calcination at 600 °C for 4 h with a heating ramp of 5 deg/min.

## 3. Results and Discussion

### 3.1. Products Characterization

#### 3.1.1. Raw and Treated Materials

Initial characterization of raw and treated Hassawi rice husk and straw samples with FT-IR spectroscopy are illustrated in Figure 1a,b, respectively. It is clear from Figure 1 that both raw straw and husk samples have identical FT-IR spectra, as well as NaOH treated ones, which means that they have the same functional groups. As a result, this observation directed us to mix both parts of the waste together to form one homogenous sample. The mixed sample was then treated with NaOH alkaline solution in order to extract and precipitate silica/silicate from the filtrate liquor by lowering its pH to 7.

#### 3.1.2. Intermediate Material

Separated solid material at pH = 7 (expected to contain silica/silicate mixture) was dried at 105 °C for 8 h. Dried material was further characterized using SEM imaging and FT-IR and XRD spectroscopic techniques. Figure 2a represents the SEM image for the extracted mixture, which shows a heterogeneous amorphous surface with a non-specific crystal structure. FT-IR spectrum (Figure 2b) shows a complete similarity between un-calcined silica/silicate extracts of mixed rice agriculture waste and that in the literature [36], where the asymmetric vibration peaks of Si–O–Si bonds appear at 1062–1080 cm^−1^ and symmetric stretching of the Si–O–Si bond was located at 801 cm^−1^.

The stretching vibration peak of O–H at the silanol groups (Si–OH) was found at 3350 cm^−1^, the bending peak at 1651 cm^−1^ corresponds to the adsorbed water H–OH bond, while the –OH stretching vibration peak is located at 2933 cm^−1^. XRD diffractogram of extracted silica/silicate mixture (Figure 2c) showed a broadened peak at 2θ = 20.44–24.50° corresponds to the silica amorphous phase. Several other crystalline peaks located at 2θ = 28.48°, 40.68°, 50.36°, 58.94°, 66.56°, and 73.84° correspond to crystalline silicate particles [37].

#### 3.1.3. Final Porous Silica Nanoparticles (MSN)

Figure 3a shows the XRD diffractogram of the fabricated amorphous silica. XRD analysis showed one broad peak centered at 23° 2θ. This peak can be attributed to the amorphous silica environment as reported by Adam et al. [38]. More importantly, no peaks for the sodium silicate could be detected as an indication of the successful preparation technique.

The FTIR-spectrum of the fabricated amorphous silica is shown in Figure 3b. The spectrum is dominated by three ideal distinct peaks of silica environment at 443.5, 801.4, 1080 cm^−1^ that indicated the symmetric vibrations of bending motion of Si–O–Si and O–Si–O groups; the symmetric stretching vibrations mode of Si–O–Si or vibrational of ring structures and the bridging oxygen atom between tetrahedral units; and the stretching vibration mode of Si–O–H asymmetric surface group, respectively. In addition, a slight small shoulder at 1200 cm^−1^ indicated the amorphous skeleton of silica [39].

Figure 3c shows the N_2_ isotherm of the fabricated silica, the isotherm exhibited type IV(a) isotherm, according to the IUPAC classification [40], as an indication for the mesoporous character of the pore system. Moreover, the pore size distribution exhibited maxima at 4.8 nm (Figure 3d), which agrees with the N_2_ sorption isotherm as an indication for the mesopore system of the fabricated silica. The texture properties of the fabricated amorphous silica were calculated from the N_2_ sorption analysis. The obtained surface area was 580 m^2^/g, the pore volume was 0.4 cm^3^/g and the pore diameter was 4.8 nm.

The morphological structure of the fabricated porous silica was studied by SEM analysis (Figure 4a). The SEM micrographs showed that the fabricated materials have sphere like-structure with a dimension between 0.35 and 45 μm. The EDX analysis (Figure 4b) showed that only two elements are present, the Si and the O, as an indication of the high purity of the prepared material. The data obtained by SEM was confirmed by HR-TEM analysis (Figure 4c,d); only sphere-like particles were obtained. More importantly, HR-TEM micrographs clearly show the porous character of the fabricated material. The obtained characterization data confirmed the possibility of fabricating amorphous mesoporous bio-based silica from sodium silicate.

#### 3.1.4. Lignocellulose

Figure 5 illustrates the characterization data for lignocellulose separated after the alkaline then acidic treatment of Hassawi rice biomass waste. SEM imaging (Figure 5a) indicated that NaOH treated biomass acquired smooth surfaces with a flake-like structure. FTIR analysis (Figure 5b) proved that the separated lignocellulosic material consisted of lignin, cellulose, as well as hemicellulose [41]. The band located at 1032 cm^−1^ corresponds to vibrational stretching of C–O in cellulose. O–H bond stretching of cellulose appeared at 3317 cm^−1^, while functional groups of lignin (aromatic rings, C=O, OH, OCH_3_) have characteristic peaks at 1509 and 1602 cm^−1^ [42]. C–H asymmetric deformation of methylene and methoxy groups’ peak is located at 2944 cm^−1^. Skeletal vibration of the lignin aromatic rings showed a peak at 1420 cm^−1^ representing the compound backbone structure.

XRD pattern for lignocellulose (Figure 5c) indicates that the separated material consists of mixed amorphous phases of silica and lignocellulose. No crystalline peaks appear in the diffractogram, which confirm the removal of cilicate phase and the presence of major amorphous lignocellulose with minor amorphous silica phases.

#### 3.1.5. Activated Carbon

The XRD diffractogram of the produced activated carbon sample (Figure 6a) does not exhibit a horizontal basic line. This observation indicates that the major parts of the matter are in the amorphous phase.

FT-IR spectrum of activated carbon produced from rice husk/straw (Figure 6b) indicates a strong inter/intra-hydrogen bonded (O–H) stretching absorption at 3400 cm^−1^, found in celluloses; a series of peaks in the fingerprint region (1800 and 600 cm^−1^), including unconjugated C=O of hemicellulose at 1746 cm^−1^; C–H deformation in lignin and carbohydrates at 1443 cm^−1^; C–H deformation in cellulose and hemicellulose at 1378 cm^−1^; C–H vibration in cellulose at 1320 cm^−1^; syringe ring and C–O stretch in lignin at 1249 cm^−1^; C–O–C vibration in cellulose and hemicellulose at 1156 cm^−1^; C–O stretch in cellulose and hemicellulose at 1083 cm^−1^; and C–H deformation in cellulose at 871 cm^−1^.

The N_2_ sorption isotherm of activated carbon is shown in Figure 6c. The isotherm is a mixture of type II and type IV(a) according to the IUPAC classification [40]. The first section of the isotherm (at low relative pressure) can be attributed to the micropore filling; moreover, the small curvature of the plateau can be explained by the multilayer adsorption of nitrogen gas molecules on the external surface. The hysteresis loop is located between relative pressures from 0.4 to 1, which indicates the presence of larger pores. The BET surface area of the sample, as calculated from the N_2_ sorption calculations, was found to be 472.59 m^2^/g.

From the SEM analysis for the physically-activated carbon samples (Figure 6d,e), it is noticeable that heterogeneous surfaces with mostly spherical particles were obtained, with particle diameters ranging from 2–5 mm. The EDX analysis (Figure 6f) was carried out on several zones of the obtained adsorbent. The results showed that the sample consisted mainly of carbon (85.23%) and a small percentage of potassium (2.44%) with an oxygen content of 12.33% appeared with a pyrolysis temperature of 900 °C.

### 3.2. Collective Products and Applications

The process of recycling Hassawi rice agricultural waste is summarized in Figure 1. All the indicated processes for final material production and applications are currently ongoing research activities by our team members. A US-Patent including the whole extraction/treatment process was registered at the United States Patent and Trademark Office [43]. After alkaline treatment of the mixed waste, cellulosic solid residue undergoes an enzymatic fermentation process in order to produce bioethanol. The same solid residue is involved in a carbonization process to produce activated carbon that is used in water treatment processes. Residue produced from rice straw is used to produce pulp, which is used efficiently in the paper industry.

The alkaline filtrate solution is treated with concentrated HCl to bring its pH down to 7, where the silica silicate mixture is precipitated and filtered off. The liquid neutral filtrate is further treated with acid (concentrated HCl) to lower its pH to 1, where lignocellulose is precipitated and separated, and is used in anticancer studies and in the pharmaceutical industry. Separated lignocellulose is also carbonized to produce graphite, which is then converted into graphene oxide by Hummer’s method, which is used in supercapacitors. Separated silica/silicate is converted into porous silica nanoparticles using pluronic copolymers in acidic media. Produced silica is then applied in the fields of anticancer and water treatment/desalination.

## 4. Conclusions

Hassawi rice husk and straw agricultural waste is simply and successfully recycled using alkaline, followed acidic chemical treatment processes. The treatment processes resulted in a variety of important natural chemical species that are applied in different crucial fields. Produced natural materials and their application fields include lignin for bioethanol production, activated carbon for water treatment processes, pulp for paper industry, lignocellulose for anticancer studies and the pharmaceutical industry, graphene oxide for supercapacitors, and silica for anticancer studies and water treatment/desalination processes. The introduced recycling procedure can be considered as a simple chemical treatment process through which a serious environmental problems such as the burning of rice biomass waste that produces hazardous emissions and increases global warming, is solved and many beneficiary natural materials and final products are obtained in turn.

## Data Availability

Not applicable.

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
