# Peer review of "Valorization of Rice Husk and Straw Agriculture Wastes of Eastern Saudi Arabia: Production of Bio-Based Silica, Lignocellulose, and Activated Carbon"

_materials, 2022, doi:10.3390/ma15113746_

Round 1

Reviewer 1 Report

Manuscript deals with reuse/recycling of rice husk and straw from Saudi Arabia. This is a short description of waste treatment  of rice husks and characterization of the obtained products. Text doesn't bring any unknown facts other then application of well established methods on a particular species in a particular geographical region. Otherwise, text is clear and concise on what it studies. Paper is well written and the text is easy to follow. Preparation procedure is well described and characterized. Conclusions are based on the data provided and are in line with the current literature.

line 179. spectrogram is not used for XRD. Diffractogram is the correct term.

line 310. "few amount" ... amount is not used correctly... Maybe write: few percent... or some potassium...

line 355. I wouldn't call it "green"... when you use strong alkali and high temperatures in the recycling process. Maybe some other word would be better suited to describe the processes.

Reviewer 2 Report

An interesting article dealing with the development of novel simple low level treatment methods  for converting rice husks and straw agricultural wastes into activated carbon and silica which can then be subsequently used for treatment of waste water and anti-cancer products.

A useful process for the technologies in the regions of concern with the main outcome and originality being the chemical treatment processes utilised.

The processes appear to be highly viable and the analysis and methodologies underpin the identity, outcomes and activity of the final products.

The article is well presented and readable, well written and scientifically sound. The analysis by SEM and XRD coupled with FTIR acurately determine the types of products proposed for both rice husks and rice straw.

A useful and valuable proposal for recycling waste products.

Reviewer 3 Report

The presented manuscript includes the study of the valorization of rice husk and straw agriculture wastes of western saudi Arabia: production of bio-based Silica, lignocellulose, and activated carbon. 
The results of the work are presented on a good level and well written, but, some general corrections should be mentioned. 
1. Please split subsection 2.1 into different subsections: separate Materials and reagents, separately equipment. Solutions move to treatment procedures.
2. Please mention all used reagents in subsection 2.1 Materials and reagents
3. In Fig 1 please show corresponding wavenumbers for discussed above functional groups with the aim to increase visibility and understandability. Please do the same for all spectra (for both FTIR and XRD) shown in the paper.
4.  Please also mention the corresponding card number for shown XRD fig2
5. XRD is a difractogram. Please change the title of Fig.2 and whole the text
6. Please do the same treatment with Fig 3
7. Fig. 3 is incorrectly classified as a Type IV isotherm. I mean that type 4 is split into 2 different subtypes. Please have a look at the IUPAC recommendations for describing isotherms (Pure Appl. Chem. 2015, 87, 1051 DOI:10.1515/pac-2014-1117) and revise the classification. 
Please, correct in different parts of the text body.
8. Please, also, compare your results with published ones.
9.  

Round 2

Reviewer 3 Report

thank you for the corrections made